# A Review of Recent Research and Application Progress in Screw Machines

**Chuang Wang * , Bingqi Wang, Mingkun Liu and Ziwen Xing**

School of Energy and Power Engineering, Xi'an Jiaotong University, Xi'an 710049, China;
starcraft@stu.xjtu.edu.cn (B.W.); mingkunliu@stu.xjtu.edu.cn (M.L.); zwxing@mail.xjtu.edu.cn (Z.X.)
* Correspondence: chuangwang@xjtu.edu.cn

**Abstract:** Screw machines, mainly including single-screw type and twin-screw type, have gone through significant development and improvement during the past decade. This paper reviews the relevant studies available in the open literature for acquiring insight into and to establish the state of the art of the research and application status of screw machines. The related research on different aspects, which would affect the performance and reliability of screw machines includes rotor profile and geometric characteristics, thermodynamic modelling, vibration and noise, lubrication and wear, control of capacity and built-in volume ratio, and liquid injection technology. In the aspect of thermodynamic modelling, the available methods, i.e., empirical or semi-empirical model, lump model, and 3D CFD model, adopted for the performance prediction and optimal design of screw machines are summarized. Then, the review covers the application status of screw machines in the fields of air compression and expansion, refrigeration and heat pump, organic Rankine cycle (ORC), and other popular applications, with an emphasis on the reported performance and progress in technologies of screw machines. Finally, conclusions and perspectives for future research in the area of screw machines are presented. The review provides readers with a good understanding of the research focus and progress in the field of screw machines.

**Keywords:** single screw; twin screw; progress; research aspects; application



## 1. Introduction

The increase in both global energy demand and environmental concerns has become the growing threat for the survival and development of mankind, pointing to the urgent need for achieving carbon peak and carbon neutrality [1]. One of the significant approaches is to improve the energy efficiency of the energy-using equipment, and the high demands on the energy efficiency can be found in regulations of EU [2], USA [3], China [4], etc. Due to the widespread use and high-power consumption of positive displacement compressors and expanders, a small improvement in the efficiency has a considerable impact on total energy use, which would contribute to decarbonization. Among them, screw machines have outstanding features and thus are widely used in small- and medium-scale gas compression and expansion applications. Screw machines mainly include the single-screw type and twin-screw type, whose typical structures are shown in Figure 1. Twin-screw machines have occupied a large market share in industrial positive displacement machines due to their high reliability, high efficiency, easy operation, well adaptability, and low maintenance costs [5]. Meanwhile, single-screw machines have exhibited sustained growth of market share in several applications because of the balanced loading of the main screw, high volumetric efficiency, low noise, and low vibration [6]. Considering the different applications of gas compression and expansion, there are four kinds of screw machines: twin-screw compressor (TSC), twin-screw expander (TSE), single-screw compressor (SSC), and single-screw expander (SSE).

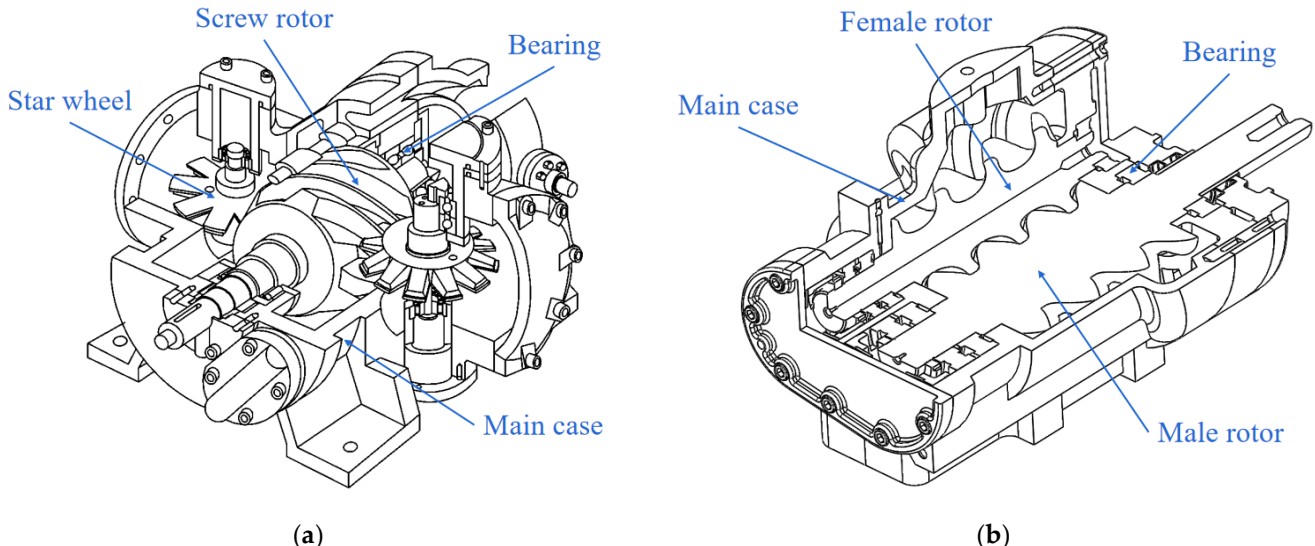

**Figure 1.** Basic structure of typical screw machines. (**a**) Single-screw machine. (**b**) Twin-screw machine.

Screw machines have gone through significant development and improvement during the past decade, which is visible in the open literature. Among them, there are four relevant review papers about screw machines. Patel and Lakhera [5] reviewed the various experimental investigations on TSCs, related to various parameters such as leakages, pulsations, noise, liquid injection, capacity control, pressure losses, optimization of rotors, indicator diagrams, etc. Wei et al. [7] summarized the research status of the leakage, rotor geometry, sealing and lubrication, processing and manufacturing of TSEs, and introduced the application status and potential utilization of TSEs. With respect to single screw machines, Wu and Zhang [6] gave a comprehensive review of the progress and development of SSC technologies, from the perspective of the current situation of SSC products, the design and processing of meshing pairs, the working process, and its performance in new applications. Ziviani et al. [8] focused on the geometry modeling of single-screw machines with emphasis on expanders.

As the similar positive displacement machines with screw rotors, single and twin-screw machines have certain similarities in the research and applications, while the aforementioned four review papers only focus on one type of screw machines from a certain angle. In contrast, this paper aims to present a comprehensive review of the relevant research papers for acquiring insight into the research aspects and application status of all screw machines, with emphasis on the similarities and differences of single and twin-screw machines. The contents would provide readers with a better understanding of the research focus and progress in the field of screw machines over the last decade.

## 2. Research Aspects of Screw Machines

### 2.1. Profile and Geometric Characteristics

Profile plays an important fundamental role in the reduction of leakage, friction, wear, and thus determines the performance and reliability of screw machines fundamentally. The wear of star wheels is a critical problem for single screw machines, and researchers from Xi'an Jiaotong University proposed a new design with optimizing new profiles from two angles: (i) dispersing wear and making contact line move continuously [9]; (ii) suspension meshing by obtaining the almost equal liquid-film forces on both flanks of the star-wheel tooth [10,11]. When designing new profiles, the corresponding manufacturing method must be considered simultaneously, aiming to achieve higher manufacturing efficiency or avoid clearance and interference between the tooth tip and the groove bottom [11,12]. Recently, they recommended the self-developed multi-column envelope profile and investigated its characteristics from aspects of geometric characteristics [13], leakage [14], heat transfer [15], liquid injection [16], thermodynamic performance [17], lubrication [18,19], and wear [20].

So far, most of the available single-screw machines employ a 6 flutes main rotor meshing 11 teeth gate rotors. However, Dhunput et al. [21] presented that the 3–10 geometry offered better isentropic efficiency compared to the 6-11 geometry especially at higher rotating speed owing to its larger discharge port by testing a semi-hermetic SSC prototype. Based on the dynamic model, Peng et al. [22] proposed that an odd-grooves screw rotor structure with a ratio of 5:9 could reduce the gas-induced torque fluctuation ratio of SSC from 0.31 to 0.15, which ensures the long-term stable operation of the motor for high pressure ratio application.

Different from single-screw machines, the profile design for twin-screw machines has been better established at the end of the 20th century, and most well-known enterprises and research institutions have developed their own profiles, such as SRM 'D' profile, Fusheng profile, Compair profile, 'N' profile and so on [23]. A new tendency is to use splines, Bezier or Non-Rational Uniform Rational B-Spline curves to represent traditional analytical curves in designing profiles, which are advantageous for curve formation or representation but inconvenient for calculating contact points of conjugate surfaces [24]. Some graphic technologies are also applied to generate profiles of rotors and grinding wheels, such as edge detection method [25,26] and pixel solution [27]. Recently, more work is focused on the optimization of the existing profiles for different applications and operating conditions based on geometric characteristics [28] or comprehensive thermodynamic simulation [29,30]. To simplify the optimization process, Hauser and Brümmer [31] proposed a method of profile optimization purely via evaluating gap conditions of TSCs instead of the thermodynamic performance. In addition to these, Ji et al. [32] presented a double measurement method based on reverse engineering to measure an unknown rotor profile of a TSC accurately by achieving the 3D radius compensation.

Industrial TSCs are usually built with constant rotor lead (Figure 2a). However, non-constant rotor lead has been proposed by some researchers for air and refrigeration TSCs, and was treated as a very potential technology for providing a larger discharge area and thus the higher isentropic efficiency by theoretical analysis [33–37], but the processing is still a difficulty. Non-constant rotor lead can be classified into two types: multiple lead (Figure 2b) and continuously variable lead (Figure 2c). The dual lead rotors are the easiest to be manufactured and TU Dortmund University successfully produced dual lead rotor prototypes by milling on a five-axis milling machine [34], but it is a pity that no experimental data was provided. Gardner Denver, Inc. [38] has introduced the continuously variable lead rotor into the twin screw cyclo-blower product since 2018, and declared up to 35% saving on energy costs. Hoang et al. [39] proposed a general method to hone screw rotors using the internal-meshing wheel, which can be used for honing screw rotors with a variable lead.

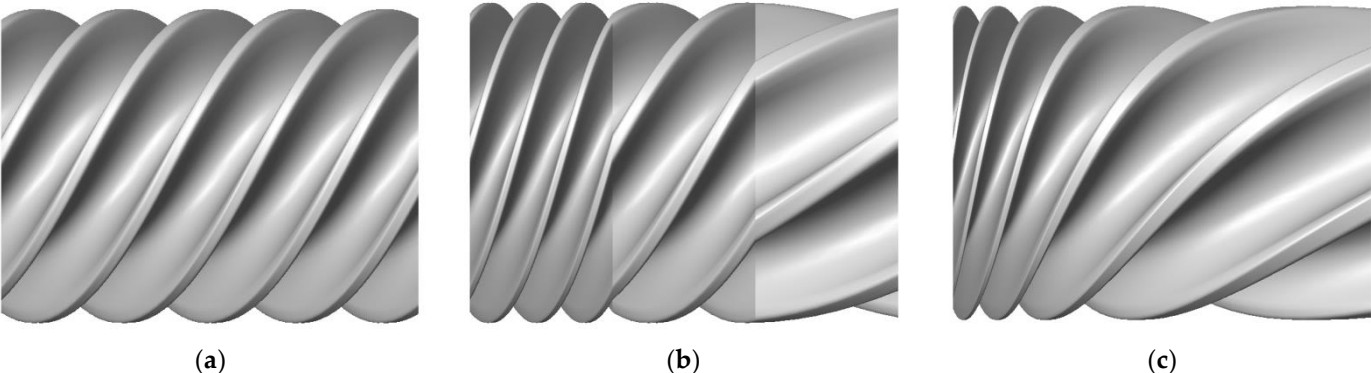

(**a**)          (**b**)          (**c**)

**Figure 2.** Rotor structure with various lead. (**a**) Constant lead (**b**) Multiple (triple) lead (**c**) Continuously variable lead.

Clearance setting has a significant effect on the efficiency and wear of screw machines. Larger clearance would cause serious degradation in volumetric efficiency and adiabatic efficiency of screw machines, due to internal leakage [40,41]. In order to evaluate the meshing clearance for twin-screw rotors, Wu et al. [42] proposed a numerical method by calculating the distance between two normal racks generated from measured discrete tooth profile points. The thermal deformation is the principal factor affecting the actual clearance distribution relative to force deformation, and thus should be considered in the design. Buckney et al. [43] predicted the possible variation of original and revised meshing clearance distribution in an oil-injected TSC, considering the effect of thermal distortions. By using a finite-element-based thermo-mechanical coupled model based on measured thermal boundary conditions, Meng et al. [44] found that the spatial position meshing error caused by the thermal deformation was one of the major reasons for the wear of the meshing pair of the SSC. The clearances were further modified based on the thermal deformation in a SSC with a capacity of 6 m$^3$/min and the results showed that the modified compressor operated reliably. Zhang and Wu [45] calculated the thermal deformation of the meshing pairs in a SSC by using the same methods, and pointed out the deformation laws of the meshing pairs and torsional deformation of the screw rotor which cause the misaligned engagement. In order to solve the problem of the gap variation of the meshing pairs, Wang et al. [46] proposed a new hot-state machining method to study the deformation characteristics of the screw rotor in the hot-state machining process, which provided some reference data for the pre-deformation of the rotor in the machining process.

*2.2. Thermodynamic Modelling*

Thermodynamic modelling is an effective method for predicting and improving the thermodynamic performance of screw machines. The modelling approaches are mainly based on three kinds of models: empirical or semi-empirical model, lump model, and 3D CFD model.

2.2.1. Empirical and Semi-Empirical Model

The empirical model doesn't describe the thermodynamic process but characterizes the machine performance through its isentropic and volumetric efficiencies by means of constant empirical values or polynomial regressions. The semi-empirical model describes the most influential phenomena by a limited number of physically meaningful equations, and correlates the important running conditions as well as design parameters of machines while other unimportant parameters are ignored, as Figure 3 shows. The empirical model and semi-empirical model have fast calculation speed and enough accuracy if verified by experiments, and thus are extensively used for system simulation, machine selection, and operation control. Krichel and Sawodny [47] applied an empirical model with the real polytropic coefficient of an oil-injected TSC into an air compressor block, in order to study the dynamics performance of the compressor. Liu et al. [48] developed a semi-empirical model of a TSC for refrigeration system simulation considering vapor injection and part-load conditions. Giuffrida [49] also proposed a semi-empirical model of an open-drive refrigeration TSC, but did not consider the presence of the lubricating oil. Later, Giuffrida [50] again revised a semi-empirical model of a SSE with particular attention paid to the mechanical losses at the shaft and the ambient heat losses.

Apart from the above traditional empirical and semi-empirical models, recent neural network methods have achieved excellent results in many applications, and thus are applied in screw machines and related systems. Alonso et al. [51] proposed a data fusion approach based on deep neural networks for the online monitoring of working conditions corresponding to the capacity control system of the TSCs in chillers. Tian et al. [52] applied the back propagation neural network model trained by BigData to predict the transient coefficient of performance of an on-site screw chiller applied in cinema, whose error was almost within ±5.0%. They [53] further presented the least square support vector machine model with a genetic algorithm for high precision prediction and control of a

screw chiller. Ping et al. [54] applied a combined back propagation neural network with a genetic algorithm model to predict and optimize the maximum power output of a SSE in organic Rankine cycle (ORC) for diesel engine waste heat recovery.

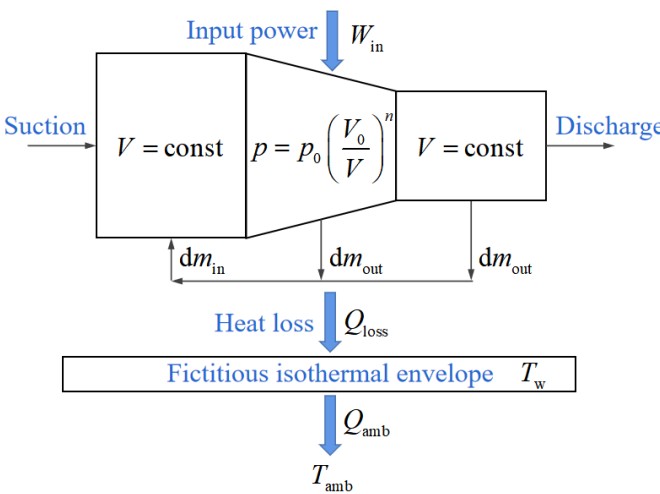

**Figure 3.** Schematic diagram of a semi-empirical model of screw machines.

2.2.2. Lump Model

As shown in Figure 4, the lump model is to describe thermophysical properties of fluids in the control volume in terms of rotating angle by developing differential control equations based on mass and energy conservation. Usually, there are two options of the lump model: $m$-$u$ model [55–59] and $p$-$T$ model [17,29,60–63]. The former is simpler in equation structures, while the latter consumes less time. Furthermore, the equations describing leakage, heat transfer, and thermophysical properties of working fluids are also needed to finish the calculation.

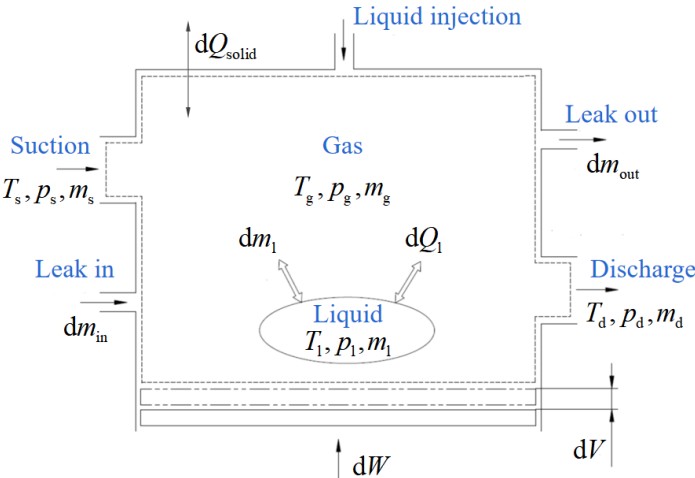

**Figure 4.** Schematic diagram of the lump model of screw machines.

Leakage is one of the key factors influencing the working process, and mainly affects the volumetric flow rate and volumetric efficiency of screw machines. There are several available leakage models used in the lump model: isentropic nozzle flow [29,55,56,60,61,63,64], orifice flow [17,55,58,61], Lin formula [14,17,55,62,65], uncompressible gap flow [14,65], incompressible and viscous pipe flow [62], adiabatic Fanno flow [59], and one-dimensional steady-state laminar flow [55,57]. Most researchers adopted the same leakage to calculate the flow rate of all leakage paths [17,29,56–58,60,61,63,64], while others adopted different leakage models for different leakage paths considering the different flow states of fluids

and the shape of leakage paths [14,55,62,65]. The flow through the suction and discharge port of screw machines can also be regarded as a kind of leakage in the modelling process, and thus its flow rate can also be calculated by the same leakage models. However, Hsieh [56] presented that the mixed fluid was discharged in the axial direction and was therefore influenced by the pressure difference across the discharge port and the axial velocity of the mixed fluid driven by the compression chamber of an oil-injected TSC. Hence, the following equation was adopted considering these two effects, where $\xi_p$ and $\xi_{sv}$ represented the relative contributions of the pressure difference and the axial velocity of the mixed fluid, respectively.

$$\dot{m}_{mix} = A_{dis} \cdot \rho_{mix} \cdot \sqrt{\xi_p \cdot \frac{2\kappa}{\rho(\kappa - 1)} \cdot \Delta p + \xi_{sv} \cdot (\omega \cdot Lead)^2} \tag{1}$$

Heat transfer is the other key factor influencing the working process, but mainly affects the shaft power of screw machines. The heat transfer within the chamber of a screw machine is often caused by the heat convection between working fluid and chamber wall, and sometimes between working fluid and injection liquid for liquid-injected screw machines. It is well known that the latter would yield much more heat transfer capacity than the former. The equation of convective heat transfer can be expressed as:

$$Q = \alpha S (T_g - T) \tag{2}$$

while $\alpha$ is the heat transfer coefficient, $S$ is the heat transfer area and $T_g$ is the temperature of the working fluid. For the heat transfer between working fluid and chamber wall, $S$ and $T$ represent the surface area and the temperature of the chamber wall, respectively [29,57,60]. Some researchers [56,58] also used $V_c^{2/3}$ to represent $S$, where $V_c$ is the volume of the leading chamber. For the heat transfer between working fluid and injection liquid, an atomization model of injection liquid is always adopted, and thus $S$ and $T$ represent the total surface area and the temperature of the atomized liquid droplets, respectively [59]. However, Wang et al. [17] thought that the injected oil cannot achieve good atomization due to the structure of the injection hole and the injection method, but would form oil film on the wall. Thus, they only considered the heat transfer between the working fluid and the wall-oil film in the calculation of an SSC. Wang et al. [55] proposed that the injected water existed in the form of both water film and water droplets, and the chamber wall was covered with water film, so they considered the heat transfer between working fluid and water film as well as water droplets, respectively, in the calculation of a water-injected TSC. Hsieh et al. [56] adopted similar assumptions to model an oil-injected TSC.

The lump model has also been introduced into some softwares for system simulations. Chamoun et al. [58] developed a lump model using Modelica to simulate a water vapor TSC for high temperature heat pump applications. Bianchi et al. [64] developed a lump model of a TSE in the commercial software GT-SUITE to investigate the performance of the expander used in the trilateral flash cycle. Kameya et al. [66] adopted the lump model of an oil-injected air TSC in the platform MATLAB/Simulink to investigate the dynamic performance of the compressor and its drive systems. In addition, on basis of the lump model, the temperature distribution of rotors [67], leakage flow rate of shaft seals [68] and gas pulsations [69] can also be further studied by supplementing some other models.

### 2.2.3. CFD Model

CFD is a powerful technology in many engineering fields, and has an increased tendency in applying to twin-screw machines. The application of CFD technologies to twin-screw machines involves unsteady flow with moving boundaries, so the high-quality grids are required to represent the highly deforming working chamber for transient simulations. As shown in Figure 5, meshing the flow field is the primary procedure in CFD modelling of screw machines, while the complex rotor shape and very small gaps in screw machines make the grid generation of the rotor filed very difficult. Researchers from City

University London made significant efforts into the rotor grid generation for twin-screw machines by means of algebraic methods [33,70–72], and have implemented it into the custom-made program called SCORG. The special rotor grid generator for rotary positive displacement machines, TwinMesh, also purposely contains the module for twin-screw machines, but can only be used in the CFD solver, ANSYS CFX. As a case, Wu et al. [73,74] applied the TwinMesh to generate the rotor grid for a refrigeration TSC. Additionally, some common commercial software, such as Gambit [75] and STAR-CCM+ [76], can also be used to generate dynamic grids for twin-screw machines, but are hardly used owing to the inconvenience.

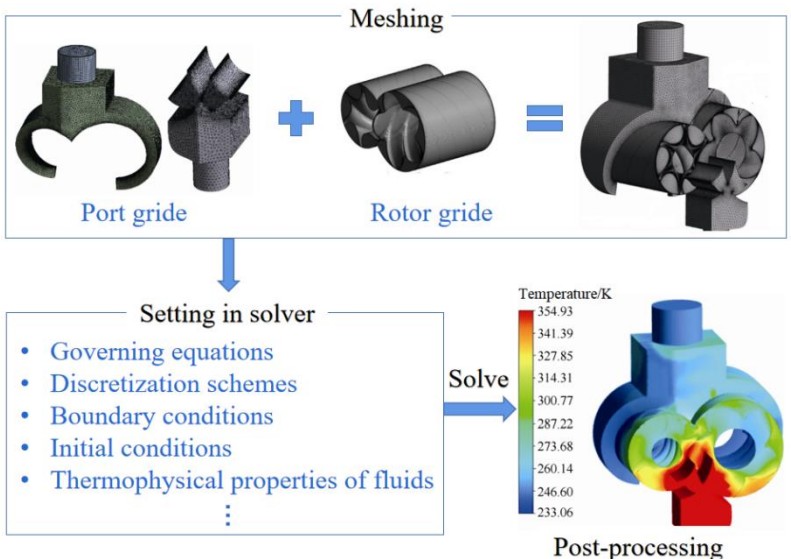

**Figure 5.** Flowchart diagram of CFD modelling of twin-screw machines.

In the CFD modelling, the choice of the CFD solver [77], the real gas equation of state [78] and the multiphase model [79] has a significant effect on calculation time and accuracy, and the Peng Robinson equation and the Volume of Fluid model are suggested. Table 1 summarizes the key information of recently reported CFD simulations of screw machines. The CFD simulation makes it possible to simulate the pressure distribution, temperature distribution, velocity distribution, and so on of the full 3D fields in space [73,75,76,80], and thus is more accurate and can provide a better understanding of the working process within twin-screw machines compared with other kinds of models. Therefore, the CFD model sometimes was used to validate the lump model [63]. Stosic [59] also pointed out that the lump model and CFD model can both obtain thermal information to determine clearance size for start-up and steady running conditions. However, the CFD model can provide a full 3D temperature field in the fluid and the solid parts, and it showed the rotor temperature was strongly uniform over the cross section and almost linear along the rotor axes. However, the accurate verification of the CFD model is still a technical issue, and the current verification method is limited to the flow visualization, and flow velocity field measurement in the suction and discharge port instead of the inner of the rotor chamber [81,82] or the comparison with the measured volumetric flow rate, shaft power, and *p-V* diagram.

As for single machines, there is little related research using the CFD model due to the lack of mature grid generation software. Liu et al. [84,85] carried out a CFD analysis of two-phase flow behavior and film thickness through a vertical helical rectangular channel for improving the efficiency of the SSE by a proper simplification. Casari et al. [83] carried out a CFD simulation of a SSE in an ORC system by using STAR-CCM+ software, neglecting the effect of the oil.

**Table 1.** Key information of recently reported CFD simulations of screw machines.

| Reference | Type and Generator of Rotor Grid | Type and Generator of Port Grid | CFD Solver | Turbulence Model | Remarks |
|---|---|---|---|---|---|
| TSC [73,74] | Structured hexahedral grids by TwinMesh | Blended tetrahedral grids and hexahedral grids by ANSYS Mesh | ANSYSY CFX | SST $k$-$\omega$ | Multiphase model: VOF |
| TSC [75] | Unstructured tetrahedral grids by Gambit and UDF | | Fluent | RNG $k$-$\varepsilon$ | - |
| TSC [76] | Blended trimmed, hexahedral, and other unstructured grids by the overlapping grid technique of STAR-CCM+ | | STAR-CCM+ | SST $k$-$\omega$ | - |
| TSC [77] | Structured hexahedral grids by SCORG | Tetrahedral grids by ANSYS Mesh | ANSYSY CFX | SST $k$-$\omega$ | Advection scheme: upwind |
| | | Body-fitted binary tree grids by Pumplinx pre-processor | Pumplinx | RNG $k$-$\varepsilon$ | |
| TSE [78] | Structured hexahedral grids by SCORG | - | ANSYSY CFX | SST $k$-$\omega$ | Advection scheme: first order upwind |
| TSC [79] | Structured hexahedral grids by SCORG | - | Fluent | SST $k$-$\omega$ | Multiphase model: Eulerian-Eulerian, VOF and mixture |
| TSC [80] | Structured hexahedral grids by SCORG | Hexahedral grids by ICEM CFD | ANSYSY CFX | SST $k$-$\omega$ | Multiphase model: Eulerian-Eulerian; advection scheme: first order upwind |
| SSE [83] | Polyhedral by STAR-CCM+ | | STAR-CCM+ | - | - |

### 2.3. Vibration and Noise

Twin screw machines would cause a substantial level of vibration and noise during operation, arising from both mechanical and fluid sources. Mujic et al. [86] reviewed the sources of noise and identified the most significant factors affecting fluid noise: the main fluid source is gas pulsation caused by unsteady flow through the suction and discharge port, and the neglectable reason is turbulent flow through the ports and clearance gaps. The parameters of injected liquid largely affected the pressure pulsation [87]. The combined working chamber model and discharge chamber model were always used by scholars to explore the influence of various parameters upon discharge gas pulsations of compressors [69,86], and the application of CFD models gained an increased tendency. In order to reduce the gas pulsation, Wu [88] applied a pressure pulsation dampener in the discharge chamber of a refrigeration TSC, leading to the decrease in vibrational acceleration of compressor under-chassis by 36.2% to 41.1% under the design frequency of 250 Hz. He et al. [89] experimentally investigated different noise control methods including a half-wavelength tube, Helmholtz resonator, and multi-cavity series muffler under different working conditions. Further comparative test results [90] indicated that the use of the exhaust muffler and the attenuation scheme on the exhaust bearing block was most effective and it reduced the compressor noise level by 8.17 dBA. Shen et al. [91] adopted two noise reduction methods including the end face attenuation passage and the discharge pipe damping for a semi-hermetic variable frequency refrigeration TSC, and achieved the mean noise reduction value in the range of 5.0 dBA–10.0 dBA when the rotational speed increased from 3000 rpm to 5100 rpm.

The main mechanical source of vibration and noise is intermittent contact between rotors for liquid-injected machines, or the contact of the synchronizing gears usually for oil-free machines. Another reason for mechanical noise was transmission error [86]. Scholars adopted different methods to analyze the conventional mechanical vibration and noise of twin screw machines, such as synthesized waveform models [92], structural finite-element along with acoustic boundary-element analysis [93], and multi-body dynamic model [94]. The other type of mechanical vibration is torsional vibration, and Willie et al. [95] found that the first torsional mode can lead to torque peak fluctuation and was manifested as

sidebands in the gear train meshing frequencies. When the second torsional frequency was identical or close to the lobe passing frequency of the rotors, it can reduce the inter-lobe clearance and lead to possible rotor-to-rotor contact.

The gas pulsation and mechanical vibration occurred in the twin screw machines would be transported outwards and cause severe vibration in the outlet piping system. Zhao et al. [96] proposed some practical measures to successfully reduce the vibration of the piping system, i.e., adding a pulsation attenuator, rearranging the pipelines, adding some pipe supports, and reinforcing the thermowells.

### 2.4. Lubrication and Wear

As mentioned in Section 2.1, researchers in Xi'an Jiaotong University tried to solve the wear of star wheels in single screw machines all the time, and thus paid much attention to the lubrication and wear characteristics of different profiles by theoretical and experimental investigations [18,19,97]. It can be found that the multi-column envelope profile had higher wear resistance and better lubrication characteristics than other profiles. Further, Wang et al. [20] carried out a theoretical research on the wear characteristics of multi-column envelope profile with varied parameters based on Hertz contact theory. Optimization analysis showed that more envelope columns with a larger center distance and larger envelope column diameter should be chosen for the multi-column envelope profile to acquire a better wear resistance.

Different from single-screw machines, the wear of meshing pairs is not a critical problem for twin-screw machines, because the gas torque acting on the female rotor can be very low by adjusting rotor profile parameters, and gears can be adopted to avoid the rotor-rotor contact under the condition of dry-running. However, it is a trend to eliminate synchromesh gears and rely on the direct contact between two rotors for oil-free twin-screw machines, especially on the condition of high-speed operation, due to much power consumed by gears. Under the circumstances, the wear of rotors becomes the key problem restricting the development of such oil-free twin-screw machines. Wang et al. [98] attempted to lighten the rotor wear in oil-free TSCs from the angle of setting the driving belt based on the contact analysis.

The lubrication characteristics of bearings are important for the stable operation of screw machines. Wang et al. [99] investigated the axis orbit of journal bearing lubricated by oil and refrigerant mixtures in a refrigeration TSC, and suggested that a small quantity of refrigerant dissolved in oil was beneficial to the stability of the rotor-bearing system and it had better choose the relatively small radius clearance and aspect ratio within a proper range. They [100] also studied the effects of rotating speed and design parameters on water-lubricated bearing characteristics in an air TSC, considering surface roughness and bending deformation of the shaft at the same time. Xie et al. [101] compared the water lubrication characteristics between two types of hydrostatic thrust bearings with different grooves to illustrate that the new structure was more suitable for SSCs. Hou et al. [102] experimentally measured the axial force on rotors of a TSC for the first time and compared the experimental results with two models, for providing a more accurate reference for the design and choice of bearings.

### 2.5. Control of Capacity and Built-In Volume Ratio

Any mismatch of capacity or pressure ratio between screw machines and systems will result in power losses and lower efficiency. Therefore, the efficient control is an important strategy to improve the performance of screw machines and the actual energy efficiency of systems. Frequency conversion has been a mature and widely used technology to control the capacity of screw machines for various applications, especially thanks to the development of permanent magnet motors, which can be seen from most experimental papers. In contrast, slide valves are usually adopted in screw machines for refrigeration applications.

Controlling the built-in volume ratio to match the operating condition of systems could avoid under-compression/expansion or over-compression/expansion power loss. Liu et al. [48] and Wang et al. [103] proposed semi-empirical models for optimizing the built-in volume ratio of TSCs with slide valves for refrigeration applications and heat pump applications, respectively. Wu et al. [104] further analyzed the relations between volume ratio, suction closure volume, discharge opening volume, and the displacement of the suction and volume ratio slide valves for a SSE used in an ORC system.

The slide valve is also normally employed in screw refrigeration machines to adjust the capacity demanded by the load. Chen et al. [105] proposed a lump model to study the working process and performance of a refrigeration TSC with a slide valve assembly under part-load conditions. It suggested that the slide valve installation angles should be as large as the installation space permits, and the fixed type slide should stop with a length of 20–25% of the rotor length when the external pressure ratio was single or else the moveable type. Wang et al. [106] adopted a similar method to investigate the effects of slide valve structure parameters on the operation process of a refrigeration SSC under part-load conditions and compared the capacity control process with the built-in volume ratio control process. Further, they [107] studied the thermal dynamic characteristics of a refrigeration SSC with the single slide valve capacity control and the frequency conversion control under part-load conditions. Results indicated that the frequency conversion control can lead to higher volumetric efficiency, lower shaft power, and higher adiabatic efficiency of the compressor than the single slide valve capacity control. Sun et al. [108] also verified that the refrigeration TSC with frequency conversion control had lower noise, better thermodynamic performance under part load conditions, higher COP, and efficiency than that with slide valve by an experimental research. Therefore, there is an obvious tendency to replace the slide valve with frequency conversion for capacity regulation.

*2.6. Liquid Injection Technology*

The liquid injection is a significant technology used in screw machines to improve their performance and reliability, and the usually injected fluids include oil, water, and refrigerant. The small circular hole is most commonly applied to inject the liquid into the suction pipe or rotor chamber of screw machines. Xie et al. [109] proposed that adopting a narrow-slit injection orifice instead of a circular hole can enhance the cooling effect based on ANSYS simulation and equivalent experiments.

Oil and water are commonly employed in screw machines for lubrication, sealing, cooling gas, and reducing noise. Graber et al. [110] investigated the influence of water and oil on the operational behavior of TSEs from the perspective of clearance sealing and frictional losses. It indicated that the water-injected expander was preferable at all speeds, but oil injection seemed to be beneficial only at low speed, mainly due to the influence of their viscosity. The study also demonstrated that the temperature-dependent dynamic viscosity of oil should be considered for reasonable modelling especially at high circumferential speeds, while the influence of water temperature can be negligible. Researchers always focused on finding the proper flow rate of injected oil or water [111–115], which depended on working conditions and the capacity of screw machines. The consistent findings were that a certain amount of oil or water can effectively improve the performance of screw machines, but further increase in the injection flow rate had little or even bad effect, and thus the optimal flow rate or injection orifice size existed. Moreover, the liquid-gas ratio of water-injected screw machines was always much larger than that of oil-injected type due to the bad sealing of water compared with oil. Further, Wang et al. [114] pointed out that proper injection mode would yield better effects of efficiency improvement of a TSC by water injection. For example, the best injection mode at 3000 rpm was to inject water of 4 L/min through port #1 located at the bottom of the casing facing the intersection of two rotor holes, and the rest of water was injected through port #2 positioned in the middle section of the rotor chamber facing two rotors. Wu et al. [116] presented that a proper injection position on the compression chamber position and low temperature oil supplied

to the discharge end bearing increased the compressor performance, while increased oil flow rate in the suction pipe and suction end bearings would decrease the TSC performance. Basha et al. [117] found out that injecting oil through two separate ports, one on each rotor, could reduce the maximum gas temperature within the chamber by 30–35 °C compared to a single port, leading to a reduction in specific power by 1.8%.

Different from oil and water, the liquid refrigerant injection is often used for reducing the extremely high discharge temperature of screw compressors, and it is always adopted along with oil injection. Zlatanovic and Rudonja [118] experimentally evaluated the effect of liquid refrigerant injection on de-superheating and oil cooling process in a two-staged ammonia refrigeration system with TSCs. Wang et al. [16] concluded that the smaller initial position and larger injection-hole diameter would improve the efficiency of a SSC. Tian et al. [119] investigated the characteristics of an ammonia TSC with liquid injection used in a two-stage compression ammonia refining system, and revealed that the mean droplet diameter of the injected liquid should be less than 100 mm to guarantee sufficient cooling effect and improve the compressor performance. Furthermore, a second liquid injection nozzle was recommended to be added and located after the start point of the compression process where the temperature glide was maximum, and the optimal flow rate was about 70% of the liquid amount injected into the suction port. Wen et al. [120] found that the injection-hole radius was the most important factor affecting the performance of the SSC and heat-pump system.

## 3. Application Status of Screw Machines

Many studies on various research aspects of screw machines have been reviewed in the previous sections. In this part, the review is focused on the application status of screw machines with an emphasis on the technology progress and reported performance in fields of air compression and expansion, refrigeration and heat pump, ORC, and other popular applications.

### 3.1. Air Compression and Expansion

In the field of air compression and expansion, a significant development of screw machines recently is the increasing application of water injection technology, which is used to lower the discharge temperature and increase the volumetric efficiency of oil-free screw air compressors [113,114,121]. Water can also be used to cool the casing of screw machines [72,122]. Moreover, Stosic [59] presented that spraying a small amount of water on the rotors can substantially decrease the maximum rotor temperature, and thus can enable oil-free screw machines to operate at a higher pressure ratio. In addition, synchromesh gears can even be deleted thanks to the development and application of the coating on rotors in oil-free twin-screw machines with water injection [114,121] or even under dry-running conditions [123]. Further, the application of water-lubricated bearings made an entirely oil-free operation successfully in the air TSC [114,121].

Twin-screw machines acquire more usage in fuel cell systems now that fuel cell technologies are paid much attention. Ous et al. [121] conducted an experimental research on the water-lubricated TSC used in a fuel cell, and water injection was also adopted to humidify the reactant air and cool the fuel cell stacks as well as the compressors. Wang et al. [122] performed a contrast experiment on three TSEs to investigate the effect of the suction port area and rotor length on the expander performance, and pointed out that increasing the flow area of the suction port, optimizing the rotor length, and reducing the leakage of expanders were crucial factors to improve the performance of the expanders. They [124] further identified the features of the working process of TSEs and pointed out that the suction throttling pressure loss and leakage showed a significant influence on the expander performance. He et al. [125] developed an air-cooled dry TSC for fuel cell systems and applied it successfully in the fuel cell system of a truck.

As for single-screw machines, the two-stage water-injected SSC unit gains popular application for polyethylene terephthalate bottle blowing system, but the stability of SSC is hard to be guaranteed in practice [6,126]. Furthermore, the scholars at Beijing University of Technology [40,60,127–132] carried out lots of experimental studies on self-developed SSEs using compered air as the working fluid. Table 2 shows the main performance parameters of screw machines for air compression and expansion applications reported in recent experimental studies.

**Table 2.** Reported performance of screw machines for air compression and expansion applications.

| Reference | Suction Pressure (Bar) | Discharge Pressure (Bar) | Rotating Speed (rpm) | Shaft Power (kW) | Volumetric Efficiency (%) | Adiabatic Efficiency (%) |
|---|---|---|---|---|---|---|
| Dry TSC [72] | 1 | 2 | 6000–8000 | - | 65–70.5 | - |
| Oil-injected TSC [111] | 1 | 8 | 1000–3000 | - | 75.5–88 | 48–70 |
| Water-injected TSC [113] | 1 | 0.6–6 | 2100–4200 | 22–80 | 55–60 | 42.5–53.5 |
| Water-injected TSC [114] | 1 | 6–9 | 2400–5400 | - | 62.5–84.2 | 56.8–74.8 |
| Dry TSC [125] | 1 | 1.4–2.2 | 5000–10,000 | 2–11 | 43–87 | 36–65.3 |
| Dry TSE [122] | 1.83–2.63 | 1 | 5000–10,000 | 2.1–2.86 | - | - |
| Dry TSE [123] | 1.4–3.0 | 1 | 1000–16,000 | 0.4–4.6 | - | 22–70 |
| Oil-injected SSE [40] | - | - | 1250–3000 | 1.2–5 | 13–66 | 14–60 |
| Oil-injected SSE [60] | 7.43–7.89 | 1.4–1.56 | 2000–3000 | 2.8–3.4 | 72.9–92.95 | - |
| Oil-injected SSE [127] | - | - | 400–3000 | 0.5–5 | - | 27–59 |
| Oil-injected SSE [128] | 6–16 | - | 1400–2800 | - | - | 48.5–65 |
| Oil-injected SSE [129] | 6.5–15 | 0.4–2.3 | 1200–3000 | 4–22 | - | 17.5–69.64 |
| Oil-injected SSE [130] | 10–50 | 1–11.1 | 1500–3000 | 6–56.55 | 38–87 | 40–63 |

*3.2. Refrigeration and Heat Pump*

As for the application of screw machines in refrigeration and heat pump, most studies are still focused on various research aspects of screw compressors, which have been reviewed in Section 2. In addition, TSCs have an increasing application in high temperature heat pump (HTHP) systems for heat recovery. Zhao et al. [133] introduced a novel HTHP system with ammonia TSCs to recover heat from condensers of refrigeration systems and supply hot water, and developed a semi-empirical model to predict the performance of the compressor and system. Further, they [134] proposed a novel refrigerant extracting TSC into a modified HTHP system adopting R245fa for heat recovery. Theoretical analysis showed that the COP of the modified system was 0.3 to 0.7 higher than that of the conventional system. Wu et al. [135] developed a capacity-regulated HTHP system with a TSC for recovering thermal energy from the waste drainage system to provide the required energy for heating processes instead of steam in the dyeing industry. It was found that that the HTHP system exhibited good system performance during the whole heating process with an average system COP of 4.2, which could save about 47% of the operating cost in comparison to the traditional steam heating. Due to the special properties of water, the water refrigeration system shows better performance at a relatively high evaporator temperature, which is ideal for HTHP applications. Chamoun et al. [58] presented a water vapor TSC used in a new HTHP, and it can achieve a high temperature lift of 40–50 K, where water was injected in the suction process of the compressor to cool the water vapor. The TSC is also a promising choice to be used in the helium refrigeration cycle to produce liquid hydrogen for hydrogen fuel cells which are paid much attention recently [136].

*3.3. ORC System*

Both kinds of screw expanders are popular for small and medium scale applications in ORC systems to recover low grade heat and waste heat. For a given screw expander, its performance is significantly affected by working conditions mainly including rotating speed [127,129,137,138] and suction pressure or pressure ratio [50,137,139]. In a commonly so-called ORC system, the suction working fluid of the expander is overheating or saturated gas state. The suction superheat has a relatively small effect on the performance of

screw expanders and systems, but there are no uniform conclusions about the influence law [50,137,140]. It is possible that the effect of suction superheat is also affected by other working conditions. Furthermore, some researchers recommended two-phase expansion to increase the efficiency of ORC systems [141]. Xia et al. [142] experimentally investigated the performance of a SSE with different suction dryness by adjusting the mass flow rate of working fluid into the evaporator. The results indicated that reducing suction dryness would reduce the power output and pressure ratio of the expander but increase the adiabatic efficiency. When the suction dryness was reduced to be zero, the suction working fluid would be in the saturated liquid state, and such a system is named as trilateral flash cycle (TFC) system. Bianchi et al. [64] presented numerical investigations of a TSE in a TFC system with different suction dryness. They found that the high density at low suction dryness allowed to intake a greater amount of mass flow rate and contributed to achieving higher indicated powers, but the best value of suction dryness was 0.1 in terms of adiabatic efficiency.

In order to improve the performance of expanders, the key is to minimize various losses, i.e., suction pressure loss, leakage loss, friction loss, and under/over-expansion loss. In contrast, heat loss was not the main factor affecting the performance of expanders [143,144]. Based on 3D CFD calculations of TSEs, Papes et al. [145] found that the biggest pressure drop was caused by a throttling loss at the suction port and therefore an optimized design of the suction port was necessary. Results also showed that the influence of the leakage flows was greater at lower rotating speed and higher-pressure ratio. Wang et al. [132] pointed out that how to enhance the pressure drop effect of leakage and reduce the flow resistance loss of inlet and exhaust passages simultaneously was an important technical premise to improve the expansion ratio by structural optimization. By optimizing the intake and exhaust structure of a SSE, Guo et al. [146] experimentally verified that the filling factor of the prototype was reduced from 125% to nearly 100%, and the highest shaft efficiency was increased from 56% to 67.7% at 3000 rpm. Zivinani et al. [147] presented that the friction losses played a major role in the total loss of a SSE followed by suction pressure drops and leakages, based on a semi-empirical model. Optimizing the built-in volume ratio is also important to improve the expander performance by reducing under-expansion or over-expansion losses [148,149]. Wu et al. [150] concluded that the optimal built-in volume ratio was not the bigger the better when SSE worked at high pressure ratio condition, because suction pressure loss also increased with the increase of built-in volume ratio. The [104] further presented that adopting slide valves can effectively improve the power output of the SSE and the net power output of the ORC system. Bianchi et al. [149]. Lei et al. [151] proposed that introducing exhaust kinetic energy utilization process in SSE can reduce under-expansion losses effectively, and the effects in large size expanders were stronger than that in small scale machines. From the angle of control, Zhang et al. [152] pointed out that increasing torque of the SSE would lead to the increase of suction pressure and thus power output as well as ORC efficiency, but the total efficiency of the expander had the optimal value. Dong et al. [153] indicated that the speed regulation of TSEs showed a great improvement potential for ORC off-design performance, and there existed two different optimal speeds to obtain maximum output power or highest thermoelectric/exergy efficiency. Moreover, Eyerer et al. [154] introduced direct liquid injection into the TSE to reduce the discharge temperature, enabling the expander to run at higher live vapor conditions and thus allowing for up to 40% higher power production depending on operating conditions.

Some researchers compared the performance of ORC systems using screw expanders with different working fluids [140,141,150,155], and the most widely used in actual applications are R123 and R245fa, as shown in Table 3. It can also be seen that the rotating speed is within 900–3000 rpm for SSEs and 1250–6000 rpm for TSEs. The maximum power output and adiabatic efficiency are 560 kW and 88% reported in the literature [137].

**Table 3.** Reported performance of screw expanders used in ORC systems in experimental studies.

| Reference | Working Fluid | Suction Pressure (Bar) | Rotating Speed (rpm) | Pressure Ratio | Power Output (kW) | Adiabatic Efficiency (%) |
|---|---|---|---|---|---|---|
| SSE [139] | R123 | 6.3–12 | 2000–3000 | 4.36–8.5 | 3–8.2 | 10–77 |
| SSE [142] | R123 | 7.6–10.4 | 900, 1200 | 2.8–4.55 | 3.8–5.12 | 39–49.5 |
| SSE [147] | R245fa | 5.66–12.3 | 2000, 3000 | 3.71–7.26 | 1.283–7.364 | 20.58–51.91 |
| | SES36 | 4.5–10.28 | | 3.63–8.83 | 0.9038–6.865 | 13.5–64.7 |
| SSE [152] | R123 | 6–12.5 | 900–2600 | 2.6–4.6 | 2–10.38 | 26–73.25 |
| SSE [156] | R123 | 12 | 3000 | 4.1–6.4 | 4.3–6.7 | 38.3–42.5 |
| SSE [157] | R123 | 6–11.5 | 2000–3000 | 4–8.5 | 2.5–8.35 | 46–73 |
| TSE [137] | R123 | 3.3–4.7 | 1250–6000 | 2.36–3.36 | 210–560 | 60–88 |
| TSE [158] | R218 | 22–32 | 1800–3000 | 1.9–2.5 | 3–20 | 17–57 |
| TSE [159] | R245fa | 4–11 | - | 2.7–6.54 | 10–51.5 | 56–70 |

*3.4. Other Popular Applications*

Mechanical vapor compression/recompression (MVC/MVR) is a very promising technology for the evaporation process and highly relies on the water vapor compression. At present, the research is mainly concentrated on the wet compression process of TSCs, which can realize higher pressure ratio and saturated temperature lift thanks to the use of water injection. Water injection can substantially reduce the power consumption as well as discharge vapor temperature [160] and even increase the vapor flow rate [161]. However, Tian et al. [161] also pointed out that too much water injection was not suggested during the actual operation of the compressor, and it was enough only if the discharged vapor can be cooled to saturation. Shen et al. [162] applied a self-developed water-injected TSC with synchromesh gears on a 50 m$^3$/day double-effect MVC desalination system, and results demonstrated that the water-injected TSC can be reliably operated in the MVC system with satisfactory system performance. Further, Shen et al. [163] again proposed theoretically a two-stage water vapor compression method using the combined centrifugal and twin-screw compressors to provide a large volume flow rate and high pressure ratio simultaneously. SSCs are also considered by some researchers for applications in MVC/MVR systems. Yang et al. [164] developed a MVR system employing a water-injected SSC and found that water injection can lead to a significant increase in evaporation capacity, pressure ratio, and specific moisture extraction rate.

Some scholars investigated the performance of screw machines used in the process of natural gas delivery and use, such as the TSE for pressure energy recovery in the pressure reduction process [165–167], the SSC for natural gas liquefaction process [112,168], and the TSC for coal seam gas compression [169]. Most of these screw machines were of the oil-injected type, while Diao et al. [165] introduced an oil-free TSE to recover pressure energy at a city gas station for the first time and its actual isentropic efficiency was about 70% at all tested operating conditions. In addition to these applications, Indice et al. [170–172] and Li et al. [173–176], and Habibi et al. [177] proposed some innovative solar electricity generation systems with TSEs using water as working fluid, but these researches are limited to theoretical analysis. Zhang et al. [178] proposed that using SSEs to perform two-phase expansion to replace the throttle valve, in order to recover the energy loss due to the throttling in the path of ammonia-lean solution in the Kalina Cycle System, and the highest cycle exergy efficiency reached 56.59%.

**4. Challenge and Recommend Future Works**

Although much work has been carried out on screw machines in recent years, a few key issues remain for further investigation, including the following aspects:

- The wear of meshing pairs is still a key issue for oil-free screw machines, especially the dry-running type without synchromesh gears, and more studies are needed to solve this problem from aspects of the profile design, coating, and materials.

- As for CFD modelling, there are few related research to single screw machines due to the lack of mature grid generators. To achieve the two-way thermal-fluid-solid interaction modelling of twin-screw machines is still intractable but necessary, as it would make the simulated results closer to real conditions, especially of ununiform clearances. Furthermore, for purpose of providing more convincing evidence to verify CFD models, much deeper and more accurate measurements of the flow and heat transfer within screw machines at the micro level are urgently recommended. The latter two works play a vital role particularly in investigating the distribution and influence mechanism of injected liquid in the working chamber of screw machines.
- A more comprehensive consideration is required in modelling and designing screw machines, for example, the comprehensive analysis of power consumptions in each position and part in screw machines, matching characteristics between screw machines and motors.
- The development and application of some advanced technologies make it successful to run the screw machines under higher efficiency, such as coating, non-constant rotor lead, refrigerant extracting, and BigData. However, these new technologies are immature and needed to be further investigated and applied to new application systems.
- Screw expanders would gain significant progress and applications under the background of carbon peak and carbon neutrality, but still are restricted to the lower efficiency compared with turbines. More efforts need to be carried out to investigate the loss mechanisms and further reduce various power losses. Furthermore, the coupling way and matching between expanders with compressors are needed to be innovated and optimized, especially for applications in fuel cell systems.

## 5. Conclusions

This paper provided a detailed review of the relevant studies on screw machines, reported in famous journals, from two perspectives of research aspects and application status. Based on the review results, some conclusions can be drawn and some perspectives are proposed as follows:

- Different research emphases were carried out by researchers on single and twin-screw machines due to their unique characteristics. Single-screw machines are still restricted by the quick wear of star wheels, which limits the rotating speed to no more than 3000 rpm. Therefore, more work is focused on designing and optimizing new profiles along with processing technologies to improve the lubrication and wear characteristics of star wheels. In contrast, the profile design for twin-screw machines has been mature, but further development of technologies to reduce vibration and noise of twin-screw machines is more urgently needed. Of course, equally important problems for both kinds of screw machines exist such as thermodynamic modelling, control methods, and liquid injection technology.
- For thermodynamic modelling, the lump model is most frequently applied to predict and improve the performance of screw machines, and has been introduced into some software for system simulations to gradually replace the empirical or semi-empirical model. The CFD model has been applied successfully in the modelling of twin-screw machines and has an increasing tendency due to the development of special grid generators, SGORG and TwinMesh, while single-screw machines are far behind in this aspect and needed to be paid more attention. However, to achieve the two-way thermal-fluid-solid interaction modelling and provide more convincing evidence to verify thermodynamic models are still key issues.
- The development and application of some advanced technologies, such as water injection, coating, materials, and water-lubricated bearings, make it successful to run the screw machines under entirely oil-free conditions even without synchromesh gears. It provides us some inspirations to develop entirely oil-free screw machines using refrigerant and water vapor as working fluid. Some other potential technologies are also proposed and even have been developed to further improve the performance

of twin-screw machines, for example, non-constant rotor lead, refrigerant extracting, and BigData. However, these new technologies are immature and needed to be further investigated.

- It is visible in recent literatures that screw compressors exhibit excellent efficiency and reliability in the new fields of HTHP, MVC/MVR, and natural gas, and there is an increasing use of screw expanders in recovering pressure energy and heat energy. However, screw expanders always present unsatisfactory performance, which is a crucial problem for their commercialization. More efforts need to be carried out to investigate the loss mechanisms and further reduce various losses, i.e., suction pressure loss, leakage loss, friction loss, and under/over-expansion loss.

**Author Contributions:** Conceptualization, C.W. and Z.X.; methodology, C.W.; formal analysis, C.W.; resources, B.W. and M.L.; data curation, B.W.; writing—original draft preparation, C.W.; writing—review and editing, Z.X.; visualization, M.L.; supervision, Z.X.; project administration, C.W. and Z.X.; funding acquisition, Z.X. All authors have read and agreed to the published version of the manuscript.

**Funding:** This research was funded by the National Natural Foundation of China, grant number 51976148.

**Institutional Review Board Statement:** Not applicable.

**Informed Consent Statement:** Not applicable.

**Data Availability Statement:** Not applicable.

**Conflicts of Interest:** The authors declare no conflict of interest.

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
