# Peer review of "A Review of Recent Research and Application Progress in Screw Machines"

_machines, doi:10.3390/machines10010062_

Round 1

Reviewer 1 Report

The review which is the subject of the paper is interesting and well organized as a whole but a number of misprints and not clear sentences are found throughout the work. Enclosed you find a detailed list of corrections suggested and questions to be answered to improve the quality of the contribution. 

Reviewer 2 Report

The review paper can be accepted for the publication provided the authors implement the ensuing requests of modifications or comments.

1. Introduction

Please refer to the diverse regulations on energy using products (EU, USA and China) which define the framework manufacturers and designers should comply with.

Please be more precise in identifying what merit the present manuscript has with respect to the review papers (4) the authors have given a reference too. The text on page 2 (lines 59 to 61) is too generic.

2. Research aspects ...

CFD model (?). The authors confuse the numerical modeling approaches (not even illustrated) with a number of commercial packages (confusing grid generators with CFD multi-purpose tool). This is not acceptable, the information readers' can be interested must refer to grid generation approaches (i.e. rotor-stator interfaces), fluid-dynamics model and numerical schemes.

3. Application

The title of the first sub-section refers to "Air compression and expansion", but example of air expanders are not considered? Is that right?

Reviewer 3 Report

  1. Line 41 and 42 should be rephrased
  2. In line 100, it should read `in order to simplify...rather than simply
  3. The review is an elaborate review work, however, I suggest that the authors include the Parameterization Techniques of screw geometries. A section should be presented on this
  4. Also, a section should be included before the conclusion on `Future research on Screw machine and applications`, for which the authors should state some expected developments/improvements that could be made 
